# Effects of Ultrasound-Guided Injection Combined with a Targeted Therapeutic Exercise in Breast Cancer Women with Subacromial Pain Syndrome: A Randomized Clinical Study

**DOI:** 10.3390/jpm12111833

**Published:** 2022-11-03

**Authors:** Lorenzo Lippi, Alessandro de Sire, Arianna Folli, Antonio Maconi, Marco Polverelli, Carlo Vecchio, Nicola Fusco, Marco Invernizzi

**Affiliations:** 1Physical and Rehabilitative Medicine, Department of Health Sciences, University of Eastern Piedmont “A. Avogadro”, 28100 Novara, Italy; lorenzolippi.mt@gmail.com (L.L.); arianna.folli23@gmail.com (A.F.); 2Dipartimento Attività Integrate Ricerca e Innovazione (DAIRI), Translational Medicine, Azienda Ospedaliera SS. Antonio e Biagio e Cesare Arrigo, 15121 Alessandria, Italy; amaconi@ospedale.al.it; 3Physical and Rehabilitative Medicine Unit, Department of Medical and Surgical Sciences, University of Catanzaro “Magna Graecia”, Viale Europa, 88100 Catanzaro, Italy; alessandro.desire@unicz.it; 4Rehabilitation Unit, Department of Rehabilitation, Azienda Ospedaliera SS. Antonio e Biagio e Cesare Arrigo, 15121 Alessandria, Italy; mpolverelli@ospedale.al.it; 5Breast Surgery Unit, Azienda Ospedaliera SS. Antonio e Biagio e Cesare Arrigo, 15121 Alessandria, Italy; cvecchio@ospedale.al.it; 6Division of Pathology, IEO, European Institute of Oncology IRCCS, Via Giuseppe Ripamonti 435, 20141 Milan, Italy; nicola.fusco@unimi.it; 7Department of Oncology and Hemato-Oncology, University of Milan, Via Festa del Perdono 7, 20122 Milan, Italy

**Keywords:** shoulder, rehabilitation, ultrasound-guided injection, cancer, exercise, quality of life

## Abstract

In this randomized controlled study, we aimed to assess the effects of US-guided injections of the subacromial bursa followed by a personalized rehabilitation program for breast cancer (BC) survivors. We assessed patients with subacromial pain syndrome without tendon lesions and with a history of post-surgical non-metastatic BC. Thirty-seven patients were enrolled and randomly assigned 1:1 to receive US-guided corticosteroid injections combined with a personalized rehabilitation program (Group A; n: 19) or US-guided corticosteroid injections alone (Group B; n: 18). The primary outcome was pain relief, assessed using a numerical pain rating scale (NPRS). The secondary outcomes were muscle strength, shoulder function, and quality of life. No major or minor late effects were reported after the multidisciplinary intervention. Statistically significant within-group differences were found in terms of NPRS (*p* ≤ 0.05) in both groups. No significant between-group differences were reported after one week. However, the between-group analysis showed significant differences (*p* ≤ 0.05) after three months of follow-up in terms of pain intensity, muscle strength, shoulder function, and quality of life. Our findings suggested positive effects of a multidisciplinary approach including US-guided corticosteroid injections combined with a personalized rehabilitation program in improving pain intensity and quality of life of BC survivors with subacromial pain syndrome.

## 1. Introduction

Recent advances in cancer treatments combined with the spreading of screening programs provided positive effects on the overall survival of patients with breast cancer (BC) [1,2,3]. As a result, BC survivors’ prevalence has been progressively increasing worldwide; however, BC survivors might often be affected by a broad spectrum of disabling cancer treatment adverse effects [4,5,6,7,8,9,10,11].

In this scenario, the scientific literature showed significant functional impairment and pain at the ipsilateral shoulder and upper limb after breast surgery and cancer treatments [12,13,14,15]. In more detail, it has been estimated that up to 64% of BC survivors might complain of functional impairment of the upper limb and significant disability after 6–36 months from surgery [12]. Several causes have been identified as potential risk factors for this condition, including prolonged immobilization, postsurgical pain, prostheses, post-mastectomy pain syndrome, Axillary Web Syndrome (AWS), Breast Cancer-Related Lymphedema (BCRL), and peripheral nerve lesions [14,16,17,18,19,20]. Altogether, these disabling sequelae might have significant implications for unilateral shoulder pain, with recent reports highlighting a high incidence of Subacromial Pain Syndrome (SPS) in BC survivors [21,22]. In particular, radiological assessments might identify several disorders on the basis of SPS in BC survivors, including supraspinatus tear (53.3%), biceps tenosynovitis (13.3%), and subdeltoid bursitis (13.3%) [21]. According to recent guidelines [23], several therapeutic interventions have been proposed, including corticosteroid injections, extracorporeal shockwave therapy (ESWT), exercise therapy, massage, and nonsteroidal anti-inflammatory drug (NSAID) [23].

On the other hand, level one evidence supports corticosteroid injections in the therapeutic management of SPS without tendon lesions, with widely documented benefits on short-term pain relief and shoulder functional improvement [23,24]. In more detail, scientific evidence level 1 supports corticosteroid injections in subacromial bursa, with evidence reporting significant advantages compared to physiotherapy in improving functioning and reducing pain [23]. In this context, an ultrasound (US)-guided procedure might provide additional benefits targeting with higher accuracy subacromial bursa commonly involved in the pathogenesis of shoulder pain, in line with the key principles of precision medicine [25].

Despite these considerations, there are still some controversies about the long-term effects of corticosteroid injections and no evidence supports the effects of corticosteroids in shoulder neuromotor control despite their crucial impact on inflammation management [15]. In contrast, a personalized rehabilitation plan might improve neuromotor control with potential benefits in the long-term management of shoulder diseases [15].

Indeed, rehabilitation has been reported as a cornerstone for BC survivors management with growing literature highlighting its role in improving physical function and overall well-being [5,26]. Moreover, rehabilitation exercises might have a role in improving shoulder kinematic and neuromotor control, targeting the multilevel etiological causes underpinning shoulder inflammation [27,28,29]. Although a multidisciplinary management might provide further advantages for BC survivors affected by SBS, there is still a gap of knowledge about the combination of these two therapeutic interventions. To date, to the best of our knowledge, the potential synergisms between these different therapeutic interventions in women with BC have not been studied yet.

Therefore, the aim of this study was to assess the effects of US-guided injections of the subacromial bursa followed by a specific rehabilitation program in terms of pain, physical function, and health-related quality of life (HR-QoL) modifications in BC survivors.

## 2. Materials and Methods

### 2.1. Participants

In this randomized controlled study, a consecutive series of patients referred to an Outpatient Rehabilitation Service in a Hospital in the Northern Italy were recruited between April 2021 and March 2022. The study protocol followed the CONSORT guidelines for randomized controlled trials [30]. Study participants were selected based on the following inclusion criteria: (a) aged over 18 years old; (b) female gender; (c) surgery for BC; (d) subacromial pain syndrome, assessed by physical examination and confirmed by US assessment; (e) shoulder pain with a Numerical Pain Rating Scale (NPRS) >5 lasting one month or longer; (f) signed informed consent.

Exclusion criteria were: (a) allergies to triamcinolone or lidocaine; (b) severe thrombocytopenia (<10,000 plt) or bleeding disorders (c) chemotherapy or radiotherapy in progress; (d) metastatic disease; (e) shoulder tendon lesions; (f) other potentials pain generators; (g) cognitive impairment or psychiatric disorders; (h) pregnancy or breastfeeding.

The patients were assessed for eligibility by an expert physician specialized in physical and rehabilitation medicine with more than 5 years of experience in cancer rehabilitation.

This study was performed in accordance with the Declaration of Helsinki [28] and was approved by the local Institutional Review Board (ASO.RRF.22.01; Prot. n. 0005266). The participants read and signed the written informed consent before being enrolled in the study. All the study participants were allowed to revoke their consent to participate in the study without any limitation.

### 2.2. Intervention

All the patients were assessed for eligibility, and patients meeting the eligibility criteria were subsequently enrolled in this study. Patients enrolled were randomly assigned by a computer-generated randomization process with a 1:1 allocation without blocks.

The randomization process was performed before baseline assessment, but both the assessors and the patients were blinded to the allocation group during the baseline (T0) assessment. Due to the intrinsic nature of the exercise intervention, blinding during the treatment and the following assessment was not possible.

Participants randomized to Group A underwent an interdisciplinary treatment including US-guided percutaneous injections in the subacromial bursa, followed by a supervised exercise program. On the other hand, Group B underwent only US-guided percutaneous injections in the subacromial bursa without any rehabilitative treatment.

In more detail, the intervention was composed as follows:–*Ultrasound-guided injections:* the procedures were performed with the patient supine and positioned with the shoulder adducted and slightly extra rotated, the upper limb laying along the side. A US machine (Mindray^®^ M7 Bio-Medical Electronics Co, Ltd., Shenzhen, China) with a linear transducer (8–12 MHz) was used to identify the subacromial bursa and to guide the injection procedure. After the preparation of the sterile field, a US-guided percutaneous injection of 1 mL of 40 mg triamcinolone acetonide combined with 3 mL of 1% lidocaine was performed in the affected subacromial bursa [31,32]. A 22-gauge needle was used and the correct positioning of the needle directly into the bursa was real-time confirmed under US guidance. Figure 1 shows the US image of the percutaneous injection procedure.

Following the injection procedure, passive mobilization of the shoulder joint was performed to promote the complete detachment of the bursa. Two injections were performed with an interval of 1 week to increase the clinical effect. All the injections were performed by the same physician with more than 15 years of experience in US-guided injection procedures. The physician was blinded to the group allocation.

–*Therapeutic exercise program:* the rehabilitation program was composed of a 1 h session performed daily for 1 week, for a total of 5 sessions in accordance with previous studies [33,34], with the aim of improve neuromotor control and patients’ education to self-training. The therapeutic exercise program started with Codman exercises, passive mobilization exercises, and active mobilization exercises in abduction, flexion, extra rotation, and intra rotation of the shoulder were performed targeting the maximum range of motion without pain. Moreover, stretching and manual therapy including myofascial release techniques were performed. This first warm-up phase lasted 20–25 min.Subsequently, active extra- and intra-rotation exercises with elastic bands were performed to improve the strength of rotator cuff. In more detail, the elastic resistance exercises were performed elastic band selected starting by a criterion of maximum repetitions. Patients were asked to perform at least 10 repetitions with a good quality of movement and without exacerbating pain with different elastic bands. The elastic bands were characterized by a color code to a specific level of resistance: yellow (thin), red (medium), green (heavy), and blue (extra-heavy). Patients performed each exercise for 3 sets of 10–15 repetitions with a 60 s rest period between each set, for a total of 4 different exercises counting 20 min of training. This exercise protocol aimed to achieve progressions during the rehabilitation program. In particular, the progression during the study program was standardized from graduating repetition counts starting from 3 sets of 10 repetitions to 3 sets of 15 repetitions. The target number of repetition was set to the maximum repetition criteria, i.e., the target the repetition was that permitted the patient to reach the point of fatigue. Exercises for the scapular stabilizing muscles were performed by scapular motions of elevation, depression, protraction, and retraction as well as joint kinesthesia and range of motion with a visual biofeedback. Lastly, proprioceptive exercises, stretching exercises, and core exercises were performed. This cooling down phase duration was 15–20 min. A therapist/patient rate of 1:1 was performed for each rehabilitation session. 

Each session was supervised by the same expert physiotherapist. The therapist that supervised the treatments and the study participants were not blinded due to the intrinsic nature of the therapeutic exercise treatment. On the other hand, all data were not analyzed by the therapist but by a blinded physician.

Each session participation was registered to assess the adherence to the supervised rehabilitation program. Participants were allowed to miss a maximum of 1 rehabilitative session. If a patient missed a session, she did one extra session in the following 3 days after the one-week exercise program. After the end of the therapeutic exercise protocol, the patients were trained and strongly encouraged to perform a home-based exercise protocol to optimize or maintain the benefits obtained from the supervised rehabilitation program. Adherence to the home-based rehabilitation program was assessed by self-reported compliance rate. Patients should have obtained a compliance rate of at least 80% or were registered as drop out.

### 2.3. Outcome Measures

All the outcome measures were assessed by the same physician blinded to the group allocation. At baseline, sociodemographic and anthropometric characteristics were collected. Primary and secondary outcomes were assessed at baseline (T0), after 1 week (T1), and after three months of follow-up (T2).

Pain intensity was considered the primary outcome of the study and was assessed by the NPRS, a unidimensional scale ranging from 0 to 10, corresponding to ‘no pain’ and ‘worst pain imaginable’, respectively.

Secondary outcomes were the following:–*Hand Grip Strength test (HGS*), using a Jamar^®^ hydraulic hand dynamometer (Sammons Preston, Rolyon, Bolingbrook, IL, USA). This outcome measure was assessed to identify the potential implications of interventions in terms of skeletal muscle strength. The measures were standardized in accordance with previous studies [5,35]. In more detail, the test was conducted three times with the participant seated on a chair. The shoulder was adducted and the elbow flexed at 90°, while the forearm was positioned in neutral, with the thumb parallel to the humerus. The wrist ulnar deviation ranges between 0° and 15° degrees and the extension between 0° and 30°. The final score was calculated by the mean of the three different measures.–*Oxford Shoulder Score (OSS)* to assess shoulder function [36]. This scale was chosen to precisely target shoulder function given that it allows a specific assessment of disability from the shoulder, and it is influenced as little as possible by other co-morbidities [37]. It included 12 items (4 items assessing pain, 8 items assessing physical function). Each item is scored on a 5-point scale ranging from 1 (minimum) to 5 (maximum). As a result, the minimum score is 12 and corresponds to no disability, while a score of 60 represents the maximum disability.–*European Organisation for Research and Treatment of Cancer Quality of Life Questionnaire (EORTC QLQ—C30)* to assess breast cancer patient’s symptoms, physical function, and quality of life [38]. The scale is composed of 30 different items. In more detail, 28 items have a score ranging from 1 to 4, and 2 items have a score ranging from 1 to 7. The items were differently combined to provide a score of the three EORTC QLQ—C30 subscales: Global health Subscale, Functional Subscale, and Symptom Subscale [39].–*Global Perceived Effect (GPE)* to assess patients’ satisfaction with the treatment performed. The scale is a 7-point Likert scale ranging from 1 (unsatisfaction) to 7 (best satisfaction) [40].

Any complications that occurred during or after the US-guided injection procedures or the rehabilitation treatment were recorded to characterize the safety.

### 2.4. Statistical Analysis

The software G*Power3 was used to calculate the sample size. The effect size was estimated by the results of Kraal et al. [41] in terms of NPRS modifications, the primary outcome of our study. Considering an alpha error of 0.05 and a power (1-Beta) of 0.90 with a two-tail Gaussian distribution, the sample size was 38 patients (19 patients in each group).

Graphpad Prism 7.0 (GraphPad Software, Inc., San Diego, CA, USA) was used to analyze the study results. Categorical variables have been represented as numbers and percentages, while continuous variables were described as means ± standard deviations. Considering the low sample size, non-Gaussian distribution of the outcome variables examined was assumed. The Shapiro–Wilk statistic was used to confirm the non-Gaussian distribution of variables. The Friedman’s Test (One-way ANOVA) followed by Dunn’s test for multiple comparisons was used to assess the differences between single variables at different time points. Due to the intrinsic nature of the GPE questionnaire, a baseline assessment was not possible and descriptive statistics have been used to present GPE intragroup results. Mann–Whitney U test was used to assess differences between groups at different time points. A *p*-value lower than 0.05 has been considered statistically significant. Descriptive statistic was used to summarize the late effects of the treatment. All the patients were considered in the statistical analysis in accordance with the initial group allocation, in accordance with the intention-to-treat principle.

## 3. Results

Fifty-one women with BC were assessed for eligibility at baseline. Out of them, three did not sign the informed consent, one was allergic to drugs used for injection, one had bleeding disorders, two had ongoing cancer therapies, five had metastatic disease, and two had SPS with tendon lesions.

As a result, 37 patients with BC were enrolled in the study and subsequently randomly assigned to Group A and Group B. Figure 2 shows in detail the randomization process in accordance with the CONSORT guidelines for randomized controlled trials [30].

As a result, Group A included 19 participants (mean age 56.05 ± 10.30 years; body mass index—BMI: 23.58 ± 2.79 kg/m^2^), while Group B included 18 participants (mean age 58.39 ± 12.09 years; BMI: 22.72 ± 3.16 kg/m^2^). The statistical analysis did not show significant differences between groups in terms of age, BMI, and cancer treatments received. Table 1 shows in detail the baseline characteristics and the between-group analysis.

The mean adherence to both supervised and home-based rehabilitation program in group A was 94%. No patient was excluded for adherence issues, and no patient was lost during the study protocol. Therefore, all the study participants were included in the statistical analysis.

Interestingly, significant within-group differences were reported in terms of shoulder pain in both groups after one week (NPRS—Group A: 8.16 ± 0.90 vs. 4.11 ± 1.73; *p* = 0.002; Group B: 7.83 ± 0.99 vs. 3.89 ± 1.37; *p* < 0.0001) and after follow-up (Group A: 8.16 ± 0.90 vs. 2.16 ± 1.39; *p* < 0.0001; Group B: 7.83 ± 0.99 vs. 4.78 ± 1.77; *p* = 0.0026). Further details are shown in Figure 3.

Moreover, the between-group analysis showed significant differences in pain intensity after the follow-up (NRS: 2.16 ± 1.39 vs. 4.78 ± 1.77; *p* < 0.0001). No significant differences in terms of pain relief were reported at T1. Figure 4 shows the results of the between-group analysis in detail.

Concurrently, appendicular muscle strength (HGST) significant within-group differences were reported in both groups after the interventions (Figure 3). Moreover, it was found significant between-group differences after 3 months of follow-up (25.11 ± 3.20 kg vs. 20.33 ± 4.92 kg; *p* = 0.0004).

Similar results were shown in OSS score and EORTC QLQ-C30 subscales. In more detail, significant within-group changes were found in shoulder function after the follow-up (OSS—Group A: 41.79 ± 5.97 vs. 17.00 ± 3.27; *p* < 0.0001; Group B: 41.61 ± 5.79 vs. 33.11 ± 6.471). Interestingly, significant between-group differences were found after the treatment (26.21 ± 5.68 vs. 32.11 ± 5.51; *p* = 0.023) and after the follow-up (17.00 ± 3.27 vs. 33.11 ± 6.47; *p* < 0.0001). See Figure 4 for further details.

EORTC QLQ-C30 functional subscale, symptoms subscale, and global health subscale significantly improved in both groups. No significant between-group differences were shown in any EORTC-QLQ-C30 subscale at T1; however, significant differences between groups were found after 3 months of follow-up for all the EORTC QLQ-C30 subscales.

Lastly, patients’ satisfaction showed significant differences between groups after the three months follow-up, with significant advantages in the intervention group (Group A: 1.79 ± 0.54 vs. Group B: 2.78 ± 1.11; *p* = 0.003).

## 4. Discussion

SPS is a widespread disorder in BC survivors affecting both physical and psychosocial well-being of these women [21]. To date, several therapeutic approaches have been proposed to treat SPS, including corticosteroid injections, ESWT, exercise therapy, and NSAID [23]. However, the optimal therapeutic approach in patients with BC is far from being fully characterized and the synergisms between multidisciplinary treatments have not been assessed. Therefore, this single-blinded randomized controlled study aimed at evaluating the role of US-guided injections in the subacromial deltoid bursa combined with rehabilitation exercises to characterize the potential synergisms between these therapeutic approaches in BC survivors.

Our data underlined significant effects in both intervention and control groups after the follow-up of 3 months, underlining that both the combined intervention and the US-guided corticosteroids injection alone significantly improve symptoms and physical function in BC survivors with SPS. On the other hand, it was reported significant advantages of US-guided bursa injections combined with a rehabilitative treatment, with significant between-group differences after the follow-up in terms of pain relief (NPRS), shoulder function (OSS), appendicular muscle strength (HST), and HR-QoL (EORTC QLQ-C30).

These data are in line with the current evidence emphasizing the need for a multidisciplinary rehabilitation intervention to precisely address this disabling issue in BC survivors. Similarly, the previous RCT by Gasparre et al. [42] reported significant effects of US-guided injection combined with home exercise program in patients with subacromial adhesive bursitis. In particular, the authors reported significant between-group differences in US-guided injection and rehabilitation exercise compared to US-guided injection alone after three months of follow-up. However, no significant effects were reported in terms of physical function [42]. In contrast, our positive results in terms of physical functioning improvement might be related to the specific exercise intervention that was tailored on BC survivor’s related conditions [26,43]. Nevertheless, these women might show a greater functional impairment of the upper limb compared to general population [12]; thus, a personalized rehabilitative treatment might target not only the functional impairment due to SPS but also the multilevel impairment related to both cancer itself and its treatments.

Although our results highlighted the positive effects of the multidisciplinary management of SPS in patients with BC, it should be noted that at 1 week after the treatment no significant between-group differences were reported in terms of pain relief. The RCT by Ellegaard et al. [44] assessed the effects of two US-guided bursa injections combined with exercise therapy in patients with SPS, failing to demonstrate a significant difference between groups after the intervention [44]. The authors concluded that exercise therapy did not improve the effectiveness of steroid injections for shoulder pain in the general population [44]. In contrast, Bennell et al. [45] reported that standardized rehabilitation programs did not provide short-term benefits in SPS compared to placebo. However, the studies reported significant benefits on long-term pain relief and physical function after a follow-up of three months [44,45].

Besides these considerations, recent guidelines for SPS management reported a level 1–2 of evidence supporting the positive effect of exercise therapy in reducing pain and improving shoulder function [23]. Moreover, exercises targeting rotator cuff and scapular stabilizers might be the most effective approach, in accordance with our study protocol [46]. In contrast, major concerns are currently reported about manual joint mobilizations [47] and myofascial release therapy [48,49]. Thus, our results emphasized the role of a personalized rehabilitation approach for patients with BC that might focus on several risk factors for upper limb functional impairment, including AWS, BCRL, and surgical scar pain [11,16,17,50,51,52,53]. Moreover, previous studies underlined that other clinical conditions altering the neuromotor control of the shoulder joint might be effectively treated by a personalized rehabilitation program [54,55,56].

The positive effects of exercise intervention are also supported by the significant results of EORTC-QLQ30. Physical exercise interventions have a widely documented positive effect on the HR-QoL of cancer patients [57,58,59]. In addition, a growing amount of evidence underlined that physical exercise improves not only physical function but also mental health in these patients [60,61]; thus, our results are in line with previous findings suggesting that physical exercise might be an effective complementary therapy to improve both physical and mental health. Accordingly, significant differences between groups in terms of GPE have also been reported, suggesting a higher level of patient satisfaction after the multidisciplinary treatment including US-guided injection and personalized rehabilitation exercises.

Taken together, our findings emphasized the positive role of a multidisciplinary rehabilitation program targeting the multilevel neuromuscular impairment of shoulder joint in patients with BC in reducing pain intensity and improving physical function and HR-QoL.

However, we are aware this study is not free from limitations: first, the study was not powered for clinically meaningful effects in different scales. Therefore, lower changes in the outcome measures might not allow us to draw strong conclusions. Moreover, a short-term follow-up did not allow us to report the effects of our treatment in the long term. Second, the lack of shoulder kinematic assessment might not provide evidence about the neurophysiological mechanisms underpinning the improvements shown by our findings; third, the lack of a third arm performing therapeutic exercise only did not allow us to understand the role of conservative management alone. However, to the best of our knowledge, this is the first single-blinded two-armed randomized clinical study assessing the role of a specific multidisciplinary treatment in BC survivors suffering from subacromial pain syndrome.

## 5. Conclusions

Although physical exercise plays a pivotal role in the complex rehabilitation framework of patients with BC, its effect on SPS in these women is still debated, and no previous RCTs have yet assessed its role combined with US-guided corticosteroid injection in terms of pain relief and physical function improvement.

Interestingly, our results suggested potential benefits of a multidisciplinary approach including US-guided corticosteroid injections combined with a personalized therapeutic exercise program. This is the first study assessing the effects of therapeutic exercise targeting the multilevel interaction between BC disabling sequelae and SPS. Further studies assessing larger samples with longer follow-ups are needed to better understand the role of rehabilitation interventions in the multidisciplinary tailored management of BC survivors affected by SPS.

## Figures and Tables

**Figure 1 jpm-12-01833-f001:**
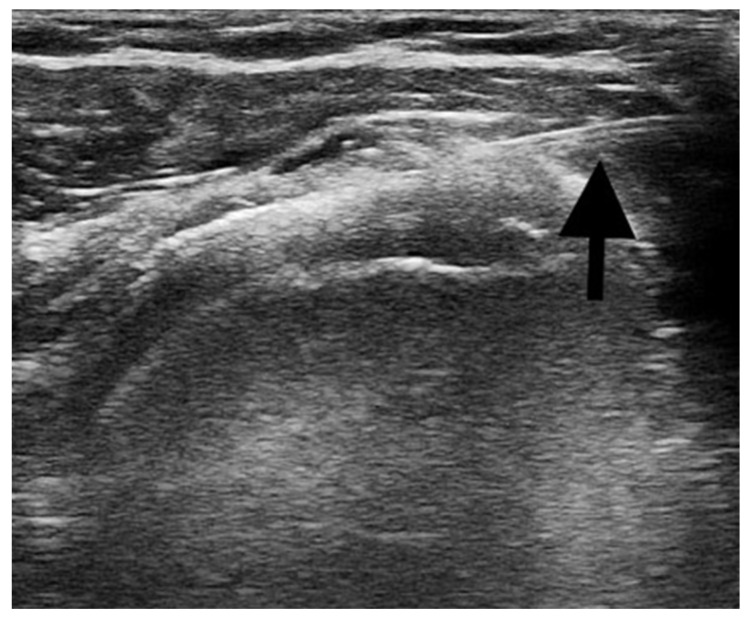
The figure shows the correct positioning of the needle (black arrow) inside the sub-acromion deltoid bursa during the ultrasound-guided injections procedure.

**Figure 2 jpm-12-01833-f002:**
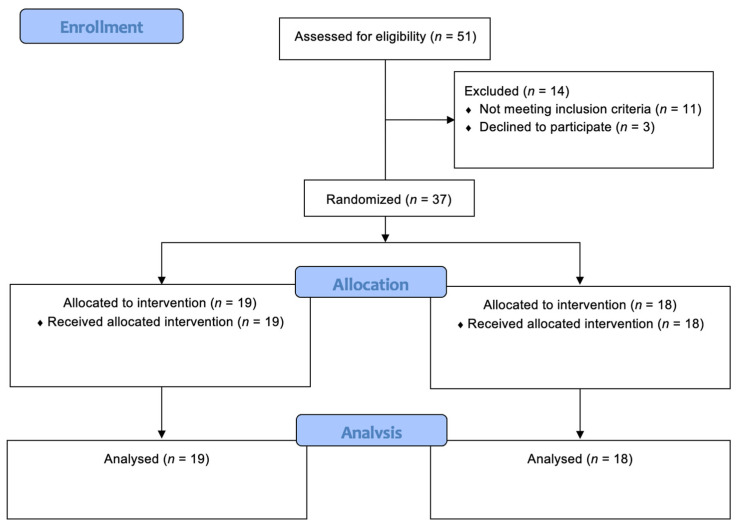
CONSORT 2010 Flow Diagram.

**Figure 3 jpm-12-01833-f003:**
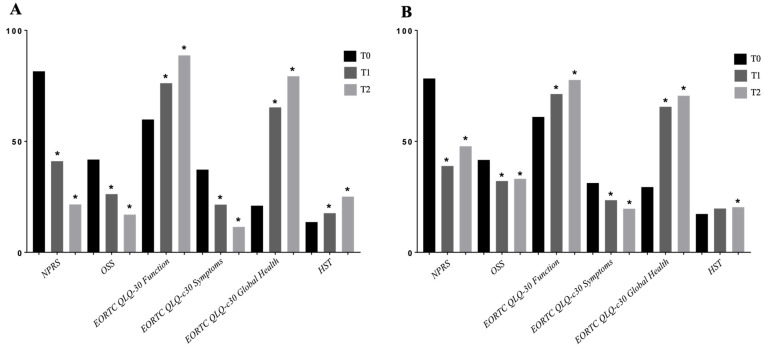
(**A**) Within-group differences in Group A. (**B**) Within-group differences in Group B. Abbreviations: * *p* < 0.05; T0: baseline; T1: after treatment; T2: after 3 months follow-up; EORTC QLQ-C30: European Organization for Research and Treatment of Cancer Quality of Life Questionnaire; HST: handgrip strength test; NPRS: Numerical Pain Rating Scale; OSS: Oxford Shoulder Score.

**Figure 4 jpm-12-01833-f004:**
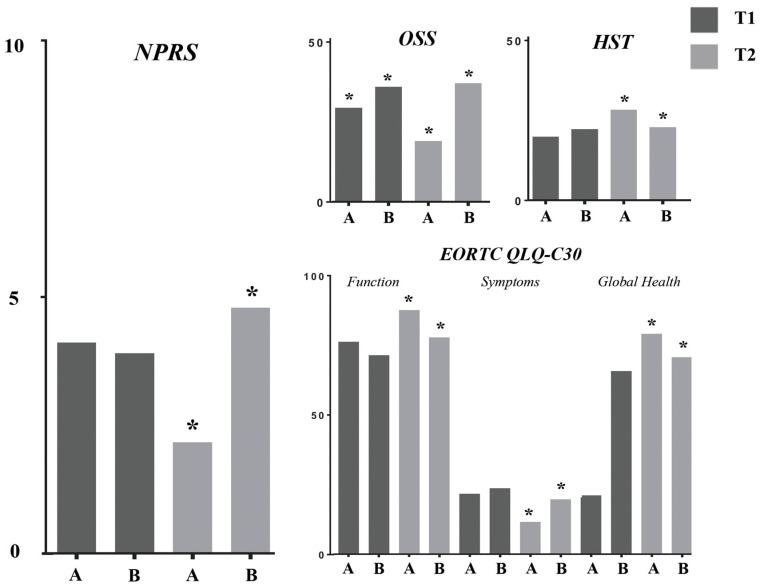
Between-group differences in the outcome measures. Abbreviations: * *p* < 0.05; T1: after treatment; T2: after 3 months follow up; A: Group A; B: Group B; EORTC QLQ-C30: European Organization for Research and Treatment of Cancer Quality of Life Questionnaire; HST: handgrip strength test; NPRS: Numerical Pain Rating Scale; OSS: Oxford Shoulder Score.

**Table 1 jpm-12-01833-t001:** Anamnestic and demographical characteristics of study population.

	Group A (n: 19)	Group B (n: 18)	*p*-Value
Age (years)	56.05 ± 10.30	58.39 ± 12.09	0.146
Sex (female)	19 (100.0)	18 (100.0)	0.999
BMI (kg/m^2^)	23.58 ± 2.79	22.72 ± 3.16	0.186
Smoker	4 (21.1)	6 (33.3)	0.999
Time from surgery (days)	674 ± 378	713 ± 322	0.174
Breast surgery			
Conservative	7 (36.8)	5 (27.8)	0.728
Mastectomy	12 (63.2)	13 (72.2)	0.728
Axillary surgery			
Sentinel lymph node	6 (31.6)	6 (33.3)	0.999
En bloc dissection	13 (68.4)	12 (66.7)	0.999
Radiotherapy	13 (68.4)	13 (72.2)	0.999
Hormone therapy	11 (57.9)	12 (66.7)	0.737
Trastuzumab	5 (26.3)	8 (44.4)	0.313
BCRL	3 (15.8)	4 (22.2)	0.693

Continuous variables are expressed as means ± standard deviations, and categorical variables are expressed as counts (percentages). Abbreviations: BMI: Body Mass Index; BCRL: Breast Cancer-Related Lymphedema.

## Data Availability

The dataset is available on request.

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
