# Peer review of "Effects of Ultrasound-Guided Injection Combined with a Targeted Therapeutic Exercise in Breast Cancer Women with Subacromial Pain Syndrome: A Randomized Clinical Study"

_jpm, 2022, doi:10.3390/jpm12111833_

Round 1

Reviewer 1 Report

General comments:

This is a well-written manuscript that is timely and addresses an issue important in cancer rehabilitation – namely, the role for bedside procedures as an adjunct to exercise or skilled therapy. The study design (RCT) is appropriate and adds badly needed high-quality evidence to a field that lacks studies like this. To that end, the authors should be commended.

I do have some concerns that I would like addressed.

Please have someone for whom English is their first language proofread this – there are some grammatical errors throughout.

I also question the intervention groups – in the introduction, you state that rehabilitation/therapy is standard of care, but then one group did not receive this. I would have liked to have seen either a third arm of just therapy/rehab, or to have the control group be only rehab. I think this limits what can be derived from the manuscript – namely, is the injection with therapy better than conservative management (physical therapy) alone.

I also question whether the personalized program – just 5 consecutive days of therapy – is enough. Improving tendinous and bursal pain generators can take months, so are we sure that this was enough?

Specific comments by section:

Introduction:

-          Would add radiation as a contributor to pain and upper extremity morbidity. This is also a good reference with additional pain generators (include as you see fit; up to you): Chang PJ, et al. A Targeted Approach to Post-Mastectomy Pain and Persistent Pain following Breast Cancer Treatment. Cancers. 2021 Oct 16;13(20):5191.

Methods and materials:

-          Please clarify whether these were inpatients or outpatients – it reads that patients were referred to a “Unit” implying these are inpatients?

-          Please clarify how patients were diagnosed with subacromial pain syndrome – physical exam? Ultrasound?

-          Would have liked to have seen “other shoulder pain generators” as an exclusion criteria. Did you exclude any patients who you evaluated, because their shoulder pain was not thought to be subacromial pain syndrome?

-          I don’t see “time since treatment completion” as being controlled for – can you comment on this? As in, were some women 5+ years out of surgery, and some only 5 weeks?

-          Can you explain why hand grip strength was measured? It’s not justified. I worry that a patient taking an aromatase inhibitor, for example, would score lower based on that. Not sure what HGS gets you. Please also clarify if AI or tamoxifen therapy was controlled for.

-          Can you justify the OSS as your patient reported outcome? Most studies I’ve seen use SPADI or Quick-DASH. This is not to say you can’t use the OSS – but please explain why this was chosen (ie its validity, etc)

Results:

-          Tables are not available for my review. I only see Figures. Please ensure that Tables are available to review in revisions (this may be for the editors)

-          Figure 4 is tough to read. The bar graphs are very close together and I am not sure which is group A and which is group B. Please rethink this or label it more clearly. I also can’t tell what is significant within group and between the groups. The text describing it is also unclear. Please explain this all much more clearly.

-          Another example is satisfaction – there was a difference between groups. Does that mean one group was not satisfied and one was? Or were both satisfied but one was more satisfied? If so, which group showed greater satisfaction?

-          I don’t see anything about compliance with ongoing home exercises – was this measured? How do we know who stopped after the 5-day physiotherapy course, and who continued with a home program?

Author Response

Dear Reviewer,

thank you for your letter and kind comments concerning our manuscript. We would like to express our sincere appreciation for your careful reviewing and invaluable comments which help us to further improve this paper.

The revisions based on your comments have been highlighted in the manuscript in yellow, and our detailed responses according to each revision are shown as followed.

Please have someone for whom English is their first language proofread this – there are some grammatical errors throughout.

We would like to thank the Reviewer for the insightful comment. Extensive English editing has been performed in accordance with the Reviewer’s suggestion.

I also question the intervention groups – in the introduction, you state that rehabilitation/therapy is standard of care, but then one group did not receive this. I would have liked to have seen either a third arm of just therapy/rehab, or to have the control group be only rehab. I think this limits what can be derived from the manuscript – namely, is the injection with therapy better than conservative management (physical therapy) alone.

We would like to thank the Reviewer for the insightful comment. We totally agree that some controversies are still open about the conventional therapies administered to BC survivors with subacromial pain syndromes. In particular, the corticosteroid injection is the most supported therapeutic intervention in current guidelines (Diercks, R.; Bron, C.; Dorrestijn, O.; Meskers, C.; Naber, R.; De Ruiter, T.; Willems, J.; Winters, J.; Van Der Woude, H.J. Guideline for diagnosis and treatment of subacromial pain syndrome. Acta Orthopaedica 2014, 85, 314-322, doi:10.3109/17453674.2014.920991.). On the other hand, rehabilitation is the most common non-pharmacological treatment proposed in BC survivors with upper limb functional impairment in the clinical setting. We better characterized this issue in the introduction section, and we improved the lack of a control group in the limitation subsection in accordance with the Reviewer’s comment.

I also question whether the personalized program – just 5 consecutive days of therapy – is enough. Improving tendinous and bursal pain generators can take months, so are we sure that this was enough?

We would like to thank the Reviewer for the insightful comment. We totally agree with the Reviewer that 5 consecutive days of therapeutic exercises might not be enough. We better characterized that patients continued home-based therapy in order to optimize or maintain the benefits obtained from the supervised rehabilitation program, in accordance with the Reviewer’s suggestion.

Specific comments by section

Introduction:

-          Would add radiation as a contributor to pain and upper extremity morbidity. This is also a good reference with additional pain generators (include as you see fit; up to you): Chang PJ, et al. A Targeted Approach to Post-Mastectomy Pain and Persistent Pain following Breast Cancer Treatment. Cancers. 2021 Oct 16;13(20):5191.

We would like to thank the Reviewer for the insightful comment. We read the manuscript suggested by the Reviewer and we improved the Introduction Section in accordance with the Reviewer’s comment.

Methods and materials:

-          Please clarify whether these were inpatients or outpatients – it reads that patients were referred to a “Unit” implying these are inpatients?

We would like to thank the Reviewer for the insightful comment. We corrected this issue in the manuscript, clarifying that the patients referred to the rehabilitation outpatient service.

-          Please clarify how patients were diagnosed with subacromial pain syndrome – physical exam? Ultrasound?

We would like to thank the Reviewer for the insightful comment. We better characterized diagnostic criteria for subacromial pain syndrome.

-          Would have liked to have seen “other shoulder pain generators” as an exclusion criteria. Did you exclude any patients who you evaluated, because their shoulder pain was not thought to be subacromial pain syndrome?

We would like to thank the Reviewer for the insightful comment. We included only patients with subacromial pain syndrome assessed by physical examination and confirmed by instrumental diagnosis. Therefore, no other pain generators were considered by our study. We further characterized eligibility criteria in accordance with the Reviewer’s comment.

-          I don’t see “time since treatment completion” as being controlled for – can you comment on this? As in, were some women 5+ years out of surgery, and some only 5 weeks?

We would like to thank the Reviewer for the insightful comment. We did not include BC patients in acute phases during active cancer treatments. We improved the inclusion criteria in accordance with the reviewer's instructions. Moreover, we better characterized the time from surgery between the two groups in Table 1, in order to better clarify that no significant differences were reported between groups at baseline assessment.

-          Can you explain why hand grip strength was measured? It’s not justified. I worry that a patient taking an aromatase inhibitor, for example, would score lower based on that. Not sure what HGS gets you. Please also clarify if AI or tamoxifen therapy was controlled for.

We would like to thank the Reviewer for the insightful comment. HGS has been assessed to underline the potential implications of rehabilitation intervention in terms of skeletal muscle strength. It is interesting to notice that to date no clear evidence is available on the impact of hormone therapy on muscle strength. On the other hand, we better characterized the study sample for the hormone therapy administered in Table 1 that has now been included in the manuscript. Moreover, we clarified that no significant between-groups differences were found in terms of hormone therapy.

-          Can you justify the OSS as your patient reported outcome? Most studies I’ve seen use SPADI or Quick-DASH. This is not to say you can’t use the OSS – but please explain why this was chosen (ie its validity, etc)

We would like to thank the Reviewer for the insightful comment. The OSS was chosen to precisely target shoulder function. While DASH or SPADI focuses also on the upper limb, OSS allows a specific assessment of disability from the shoulder and is influenced as little as possible by other co-morbidities (Younis, F., Sultan, J., Dix, S., & Hughes, P. (2011). The range of the Oxford Shoulder Score in the asymptomatic population: a marker for post-operative improvement. Annals of The Royal College of Surgeons of England, 93(8), 629–633.) including lymphedema or other common conditions in BC. We would like to thank the reviewer for his/her question that allow us to further improve the paper.

Results:

-          Tables are not available for my review. I only see Figures. Please ensure that Tables are available to review in revisions (this may be for the editors)

We would like to thank the Reviewer for the insightful comment. We included the Table in the manuscript in order to make easier its revision.

-          Figure 4 is tough to read. The bar graphs are very close together and I am not sure which is group A and which is group B. Please rethink this or label it more clearly. I also can’t tell what is significant within group and between the groups. The text describing it is also unclear. Please explain this all much more clearly.

We would like to thank the Reviewer for the insightful comment. We improved both the Figure and the Results Section in accordance with the Reviewer’s comment.

-          Another example is satisfaction – there was a difference between groups. Does that mean one group was not satisfied and one was? Or were both satisfied but one was more satisfied? If so, which group showed greater satisfaction?

We would like to thank the Reviewer for the insightful comment. Due to the intrinsic nature of the GPE questionnaire, a baseline assessment is not possible (because it represents an assessment of patient’s satisfaction with the treatment that is not yet performed). Thus, we can provide only a descriptive statistic about the GPE intragroup results. On the other hand, the between groups analysis showed that exercise therapy combined with US-guided injection provides advantages in patient satisfaction. In accordance with the Reviewer’s suggestion, we improved both Statistical Methods Section and Results Section better characterizing this issue and reporting numerical data.

-          I don’t see anything about compliance with ongoing home exercises – was this measured? How do we know who stopped after the 5-day physiotherapy course, and who continued with a home program?

We would like to thank the Reviewer for the insightful comment. Adherence to the whole rehabilitation treatment was monitored both during the supervised program and home-based program. Adherence was assessed by presence registration during the supervised training and self-reported adherence to the program at home. Patients should have obtained a compliance rate of at least 80% or were registered as drop out. We better clarified this issue in both methods and results Section in accordance with the Reviewer’s suggestion. 

Reviewer 2 Report

Dear authors

I have read the current manuscript with great interest and would like to commend you on a sound and relevant study. the impingement syndrome is a problem that many patients with breast cancer face, and it is very difficult to treat in the clinic. Hence, further advances in treatment options are welcome.

However, I have some suggestions that may add to the quality and transparrency of the reported study, listed point by point below. 

1. Title: The word "rehabilitation" in this context is too broad a term for the intervention. In the definition of rehabilitation lies many different approaches to recovering from a cancer diagnosis, such as councelling by social worker, occupational therapy, nurse intervention, dietary interventions... often a multidisciplinary approach. The intervention in this study is mainly exercise as the add on in the experiemental group, and this is the effect that you examine! Therefore, it would be more precise to use the term exercise. 

The second thing I would like to mention, is that I don't understand why you call it a pilot study. You have powered the study to show an effect of an intervention, so in my view you could call it a randomized controlled trial  and you could also state that you examine the effect, not just the efficacy, of exercise and injections. 

My recommendation would therefore be to entitle the manuscript; "Effect of ultrasound-guided injections combined with targeted exercise ...."   

2. p.2, line 46: "adverse events" is all well and good, but in my opinion you would reach a larger population of readers if you go with the more modern term "late effects" which is a whole new field in research. Quick-searches in the literature databases by clinicians might be caught by this term.

3. p. 2, lines 73-75: This is somewhat an unclear sentence to me. Are you saying that the literature suggests a role of corticosteroid injections on neuromotor control? And that this role could be mitigated by offering an individualized exercise regime? Please consider rephrasing for clarity. 

4. P. 2, line 78: "rehabilitation exercises". I think the terminology used in this manuscript is a bit problematic. If you mean "exercises" then use this term in stead fo rehabilitation, which is a term that covers much more than exercise. I commented on this allready in the title. Please consider revising throughout the manuscript for consistency and clarity in terminology. 

5. P. 2, line 84: You often use the term BC women. In many scientific communities this would be considered stigmatizing. These are WOMEN first and foremost and then they have breast cancer. So writing in stead "women with breast cancer" is the more politically correct wording. 

6. P. 2, lines 85-87: aim. Consider naming this a RCT that aims to test the effect, as mentioned in the comment on title. Otherwise, please state clearly why this is a pilot study (in the methods section). 

7. P. 2, line 95: Participants. The pain criteria, - did it have to have lasted for some time or was the momentary pain enough for a patient to be eligible?

8. P. 3, line 100-101: "eligible patients were assessed" - I think you mean that patients were assessed for eligibility by an expert... ? Please rephrase og please clarify. Also "years" is a very unprecise term. Could you specify a bit? Maybe use "more than x years.." or something other. 

9. P. 3, lines 109-111: Were patients assessed after randomization or the other way around? Consort guidelines recommend baseline assessment before randomization to prevent bias. If this did not occur, you might want to consider mentioning it as a weakness and maybe argue why you did it this way. Secondly, how can both participants and investigators be blinded, if they know what the study is about? patients must have received study information in order to give informed consent, and clinicians also...?

 10. p. 4, lines 139-156: The exercise program should be described transparently in accordance with the FITT principle (frequency, intensity, time and type) so that the exercises volume and dose (the "pill") can be precisely reproduced. Please revise this section for transparrency. Further, I am surprised that you exercise every day, as normally strengthening exercises would need a restitution day between sessions, and especially if you are an untrained individual. Could you please provide the rationale for this frequency in exercise, either in the background or the methods section, whatever you find appropriate. Line 144; you write "progression thorugh the exercsise program". Please decribe the progression protocol. 

11. P. 4, lines 147-149: this statement about blinding seems to contradict your previous statement? Please clarify.

12. p. 4, lines 151-152: do you mean that if a patients misses a session, she will do an extra session after the one-week exercise program? Please see if you can formulate this more elegantly. 

13. P. 4, line 164: does the pain have to located in the shoulder? This should be stated. 

14. p. 4, line 171: I am not sure prono-supination is a term in english? Please check. 

15. p. 5, line 191: There is no mention of providing clinically relevant or clinically meaningful effects. Often one refer to a change in NPRS of 2 points or more as being deemed clinically relevant. Seing as you haven't powered the study for a clinically meaningful effect, I would strongly recommend that you put this aspect into your discussion of the findings. 

16. p. 6, Figure 2. It seems pointless to bring categories where there are no patients, e.g. excluded from analysis (n=0). I don't recall having seen this in other publications and would recommend that you skip this. 

17. P. 7, Figure 4. In figure legends you present T0, but it doesn't appear in the figure itself, - I recommend ommitting this. Further, there are two identically coloured bars for each timepoint, but no label on this. Please look into finding a way to describe which bare is which group. 

18. It is custumary to start the discussion section with an overview of the main findings from the results. Please consider if you might find it possible to conform to this, which makes it easier for readers to orient themselves in the manuscript. 

19. P. 9, lines 334. Limitation is small sample size, but  you did actually do a power calculation and succeeded in recruiting the target number, so how come it is a limitation? 

20. P. 9, lines 339 and 349 are very much identical. Maybe this can be modified. 

21. Several English edits should be made, examples are: P.2, line 49: "more in detail" should be "in more detail". P.4, line 65; should be "short TERM pain relief". p. 4, line 93: "basing" should be "based on". p. 3, line 114: "on the contrary" - not so suited here as it is not an opposite situation for the controls, only an add-on intervention for the intervention group.

17. P. 6, Figure 3.A. The figure legends don't match the labels on the figure for Hand grip strength; HST vs HGST.

Author Response

Dear Reviewer,

thank you for your letter and kind comments concerning our manuscript. We would like to express our sincere appreciation for your careful reviewing and invaluable comments which help us to further improve this paper.

The revisions based on your comments have been highlighted in the manuscript in yellow, and our detailed responses according to each revision are shown as followed.

  1. Title: The word "rehabilitation" in this context is too broad a term for the intervention. In the definition of rehabilitation lies many different approaches to recovering from a cancer diagnosis, such as councelling by social worker, occupational therapy, nurse intervention, dietary interventions... often a multidisciplinary approach. The intervention in this study is mainly exercise as the add on in the experiemental group, and this is the effect that you examine! Therefore, it would be more precise to use the term exercise.

The second thing I would like to mention, is that I don't understand why you call it a pilot study. You have powered the study to show an effect of an intervention, so in my view you could call it a randomized controlled trial and you could also state that you examine the effect, not just the efficacy, of exercise and injections.

My recommendation would therefore be to entitle the manuscript; "Effect of ultrasound-guided injections combined with targeted exercise ...."  

We would like to thank the Reviewer for the insightful comment. In line with the Reviewer's comment, we changed the study design and the title accordingly.

  1. p.2, line 46: "adverse events" is all well and good, but in my opinion you would reach a larger population of readers if you go with the more modern term "late effects" which is a whole new field in research. Quick-searches in the literature databases by clinicians might be caught by this term.

We would like to thank the Reviewer for the insightful comment. We improved the paper in accordance with the Reviewer’s suggestion.

  1. p. 2, lines 73-75: This is somewhat an unclear sentence to me. Are you saying that the literature suggests a role of corticosteroid injections on neuromotor control? And that this role could be mitigated by offering an individualized exercise regime? Please consider rephrasing for clarity.

We would like to thank the Reviewer for the insightful comment. To the best of our knowledge, no evidence is currently available on the role of corticosteroid injections on shoulder neuromotor control. Thus, we rephrased the sentence in accordance with the Reviewer’s suggestion.

  1. P. 2, line 78: "rehabilitation exercises". I think the terminology used in this manuscript is a bit problematic. If you mean "exercises" then use this term in stead fo rehabilitation, which is a term that covers much more than exercise. I commented on this allready in the title. Please consider revising throughout the manuscript for consistency and clarity in terminology.

We would like to thank the Reviewer for the insightful comment. We changed the term “rehabilitation exercise” to “therapeutic exercise”.

  1. P. 2, line 84: You often use the term BC women. In many scientific communities this would be considered stigmatizing. These are WOMEN first and foremost and then they have breast cancer. So writing in stead "women with breast cancer" is the more politically correct wording.

We would like to thank the Reviewer for the comment. We apologize for the inconvenience. We improved all the manuscripts accordingly.

  1. P. 2, lines 85-87: aim. Consider naming this a RCT that aims to test the effect, as mentioned in the comment on title. Otherwise, please state clearly why this is a pilot study (in the methods section).

We would like to thank the Reviewer for the insightful comment. We changed the whole manuscript naming this as RCT aiming at testing the effects.

  1. P. 2, line 95: Participants. The pain criteria, - did it have to have lasted for some time or was the momentary pain enough for a patient to be eligible?

We would like to thank the Reviewer for the insightful comment. We better characterized that patients were eligible if pain lasts one month or longer.

  1. P. 3, line 100-101: "eligible patients were assessed" - I think you mean that patients were assessed for eligibility by an expert... ? Please rephrase og please clarify. Also "years" is a very unprecise term. Could you specify a bit? Maybe use "more than x years.." or something other.

We would like to thank the Reviewer for the insightful comment. We improved the sentence.

  1. P. 3, lines 109-111: Were patients assessed after randomization or the other way around? Consort guidelines recommend baseline assessment before randomization to prevent bias. If this did not occur, you might want to consider mentioning it as a weakness and maybe argue why you did it this way. Secondly, how can both participants and investigators be blinded, if they know what the study is about? patients must have received study information in order to give informed consent, and clinicians also...?

We would like to thank the reviewer for the insightful comment. Patients were assessed before randomization process only for the eligibility criteria. Randomization process was performed before baseline assessment, but both the assessors and the patients were blinded to the allocation group during the baseline assessment. We better characterized these issues in the methods section in accordance with the Reviewer’s comment. 

  1. p. 4, lines 139-156: The exercise program should be described transparently in accordance with the FITT principle (frequency, intensity, time and type) so that the exercises volume and dose (the "pill") can be precisely reproduced. Please revise this section for transparrency. Further, I am surprised that you exercise every day, as normally strengthening exercises would need a restitution day between sessions, and especially if you are an untrained individual. Could you please provide the rationale for this frequency in exercise, either in the background or the methods section, whatever you find appropriate. Line 144; you write "progression thorugh the exercsise program". Please decribe the progression protocol.

We would like to thank the reviewer for the insightful comment. We improved the characterization of the therapeutic exercise program in the methods section better characterizing the standardization of therapeutic exercises and progression performed, in accordance with the Reviewer’s comment.

  1. P. 4, lines 147-149: this statement about blinding seems to contradict your previous statement? Please clarify.

We would like to thank the reviewer for the insightful comment. We better clarified this issue in accordance with the Reviewer’s suggestion.

  1. p. 4, lines 151-152: do you mean that if a patients misses a session, she will do an extra session after the one-week exercise program? Please see if you can formulate this more elegantly.

We would like to thank the reviewer for the insightful comment. We rephrased the sentence accordingly.

  1. P. 4, line 164: does the pain have to located in the shoulder? This should be stated.

We would like to thank the reviewer for the insightful comment. We clarified that pain was located in the shoulder in accordance with the Reviewer’s suggestion.

  1. p. 4, line 171: I am not sure prono-supination is a term in english? Please check.

We would like to thank the reviewer for the insightful comment. We better characterized the forearm neutral position following the Reviewer’s suggestion.

  1. p. 5, line 191: There is no mention of providing clinically relevant or clinically meaningful effects. Often one refer to a change in NPRS of 2 points or more as being deemed clinically relevant. Seing as you haven't powered the study for a clinically meaningful effect, I would strongly recommend that you put this aspect into your discussion of the findings.

We would like to thank the reviewer for the insightful comment. We discussed this issue in the discussion section in accordance with the Reviewer’s suggestion.

  1. p. 6, Figure 2. It seems pointless to bring categories where there are no patients, e.g. excluded from analysis (n=0). I don't recall having seen this in other publications and would recommend that you skip this.

We would like to thank the reviewer for the insightful comment. We removed the categories without patients.

  1. P. 7, Figure 4. In figure legends you present T0, but it doesn't appear in the figure itself, - I recommend ommitting this. Further, there are two identically coloured bars for each timepoint, but no label on this. Please look into finding a way to describe which bare is which group.

We would like to thank the reviewer for the insightful comment. We improved the figure following the Reviewer’s instruction.

  1. It is custumary to start the discussion section with an overview of the main findings from the results. Please consider if you might find it possible to conform to this, which makes it easier for readers to orient themselves in the manuscript.

We would like to thank the reviewer for the insightful comment. We improved the second paragraph study of the Discussion Section summarizing the main results of the in accordance with the Reviewer’s suggestions.

  1. P. 9, lines 334. Limitation is small sample size, but you did actually do a power calculation and succeeded in recruiting the target number, so how come it is a limitation?

We would like to thank the reviewer for the insightful comment. We removed this limitation by the limitation subsection in accordance with the Reviewer’s comment.

  1. P. 9, lines 339 and 349 are very much identical. Maybe this can be modified.

We would like to thank the reviewer for the insightful comment. We rephrased the sentence in the Conclusion Section following the Reviewer’s suggestion.

  1. Several English edits should be made, examples are: P.2, line 49: "more in detail" should be "in more detail". P.4, line 65; should be "short TERM pain relief". p. 4, line 93: "basing" should be "based on". p. 3, line 114: "on the contrary" - not so suited here as it is not an opposite situation for the controls, only an add-on intervention for the intervention group.

We would like to thank the reviewer for the insightful comment. We improved the manuscript following the Reviewer’s suggestions.

  1. P. 6, Figure 3.A. The figure legends don't match the labels on the figure for Hand grip strength; HST vs HGST.

We would like to thank the reviewer for the insightful comment. We improved the Figure legend accordingly.

Round 2

Reviewer 1 Report

Thank you for your revisions - you have suitably responded to my concerns.